# Analysis of Olive Oil Mill Wastewater from Conventionally Farmed Olives: Chemical and Microbiological Safety and Polyphenolic Profile for Possible Use in Food Product Functionalization

**DOI:** 10.3390/foods14030449

**Published:** 2025-01-30

**Authors:** Lino Sciurba, Serena Indelicato, Raimondo Gaglio, Marcella Barbera, Francesco Paolo Marra, David Bongiorno, Salvatore Davino, Daniela Piazzese, Luca Settanni, Giuseppe Avellone

**Affiliations:** 1Department of Agricultural, Food and Forest Sciences, University of Palermo, Viale delle Scienze, Bldg. 5. 90128 Palermo, Italy; lino.sciurba@unipa.it (L.S.); raimondo.gaglio@unipa.it (R.G.); francescopaolo.marra@unipa.it (F.P.M.); salvatore.davino@unipa.it (S.D.); luca.settanni@unipa.it (L.S.); 2Department of Biological, Chemical and Pharmaceutical Science and Technology (STEBICEF), University of Palermo, Via Archirafi, 90123 Palermo, Italy; serena.indelicato@unipa.it (S.I.); david.bongiorno@unipa.it (D.B.); beppe.avellone@unipa.it (G.A.); 3Department of Earth and Marine Sciences (DiSTeM), University of Palermo, Via Archirafi, 90123 Palermo, Italy; 4Centre for Sustainability and Ecological Transition (CSTE), University of Palermo, Piazza Marina, 90133 Palermo, Italy

**Keywords:** olive oil mill wastewater, micro-contaminants, pesticides, polyphenols, lactic acid bacteria, active substances

## Abstract

This study aimed to perform an in-depth investigation of olive oil mill wastewater (OOMW). Two OOMW samples (OOMW-A and OOMW-B) from conventionally farmed olives were collected from two different olive oil mills in Palermo province (Italy). Multiresidual analysis indicated that both OOMW samples were unsuitable for food production due to pesticide residues. Specifically, OOMW-A contained 4 active compounds totaling 5.7 μg/L, while OOMW-B had 16 analytes with a total content of 65.8 μg/L. However, polyphenol analysis in the OOMW revealed 23 compounds with high concentrations of hydroxytyrosol, secoiridoid derivatives, phenolic acids, flavones, and total polyphenol content ranging from 377.5 μg/mL (for OOMW-B) to 391.8 μg/mL (for OOMW-A). The microbiological analysis of OOMW samples revealed only detectable viable bacteria (10^2^ CFU/mL) of the lactic acid bacteria (LAB) group. Two distinct LAB strains, *Lactiplantibacillus plantarum* OMW1 and *Leuconostoc mesenteroides* OMW23, were identified. These strains demonstrated notable acidification capabilities and produced antibacterial compounds. In conclusion, despite the high polyphenolic content and microbiological suitability of OOMW, the presence of micro-contaminants hinders their use in food production. Thus, further studies are underway to investigate OOMW from organically farmed olives for bakery product functionalization, employing the two selected LAB strains resistant to olive polyphenols as leavening agents.

## 1. Introduction

Agricultural waste and food by-products often contain high concentrations of bioactive compounds. These wastes not only impose disposal costs on producers but also pose environmental challenges [1]. However, recovering these bioactive compounds is essential for promoting a circular economy. Primarily, these substances can be used to enhance food products, thereby adding significant value [2]. The growing demand for functional foods has driven the food industry to develop innovative products using agricultural waste and by-products (W&BP) [3]. These materials are typically dehydrated and converted into powder before being incorporated into new food items [4,5].

Olive cultivation plays a major role in the rural economy of southern Italy [6]. In Sicily alone, the total area dedicated to olive cultivation amounted to over 15,000 hectares in 2019 [7]. The production of olive oil generates large quantities of waste, mainly pomace and olive oil mill wastewater (OOMW). The disposal of OMWW in Italy is regulated by Law 574/1996 and more recently by Art. 74 and 112 of Decree 152/2006, which allow the agronomic use of OMWW from continuous cycle plants by spreading it in the field for a maximum of 80 m^3^ ha^−1^ per year. However, the management of olive oil mill wastewater (OOMW) poses a significant challenge for olive oil-producing countries, both due to its environmental impact and the associated disposal costs. Traditionally, these residues are managed through combustion, use as fertilizers, discharge into water bodies, or storage in evaporation lagoons. However, such practices can cause severe environmental damage due to the high organic content and phytotoxicity of OOMW [8,9,10,11]. Typically, OOMW contains up to 80% water, has an acidic pH, and includes various organic compounds such as oils, fatty acids, polyphenols, sugars, and proteins. The presence of carbohydrates and proteins makes these wastes highly fermentable. Many compounds found in OOMW, including vitamins, minerals, fatty acids, dietary fiber, and antioxidants, are biologically active. Particular attention has been given to antioxidant compounds, especially the phenolic fraction [12]. Previous studies have demonstrated that OOMW-derived extracts can enrich bread and cereal-based baked goods, enhancing their polyphenol content and antioxidant capacity [13,14,15]. However, no published studies have explored the direct use of untreated OOMW as a raw material in bread production, as research has predominantly focused on the extraction of individual bioactive compounds.

In alignment with the principles of the circular economy and European initiatives aimed at waste reduction and sustainability in the food industry, this study proposes the reuse of untreated OOMW as a direct substitute for water in bread production. This research is part of a broader project aimed at creating innovative healthy food products by reusing OOMW. The challenge is to functionalize bread by incorporating bioactive compounds from OOMW while simultaneously reducing waste generation. This approach not only facilitates the development of a closed-loop production system but also holds the potential for zero waste generation, delivering significant economic and environmental benefits for the olive oil industry. Therefore, the primary objective of this study was to analyze OOMW for its potential application in food production. Specifically, the OOMW samples were examined to (i) identify and quantify phytosanitary active compounds; (ii) classify and measure polyphenols; (iii) assess the microbiological profile, including the presence and levels of spoilage and/or pathogenic microorganisms, as well as beneficial agents; and (iv) isolate, identify, and characterize lactic acid bacteria (LAB) for potential use as sourdough fermenting agents.

## 2. Materials and Methods

### 2.1. Sample Collection

OOMW samples were collected from two olive oil mills located in the province of Palermo (Italy). Specifically, OOMW-A was collected from a factory in Camporeale, while OOMW-B was from a factory in Casteldaccia. Both samples were promptly collected after milling in 10 L food-grade plastic canisters (Ecoplast S.r.l., Gela, Italy) and transported to the Food and Agricultural Microbiology laboratory at the University of Palermo (Palermo, Italy) within 6 h under refrigeration using portable coolers for immediate microbiological evaluation. Samples designated for physicochemical analysis were stored at −20 °C until testing.

### 2.2. Analysis of Micro-Contaminants

Pesticide residue was analyzed with an Ultra Performance Liquid Chromatography system (UPLC-ACQUITY I-Class, Waters, Milford, MA, USA) coupled with a mass analyzer Xevo TQ-S (Waters, Milford, MA, USA). In total, 10 mL of OOMW was centrifuged (10,000 rpm, 15 min) and then filtered (0.22 μm) and diluted 100-fold with a 1:1 water/methanol mixture. A total of 100 μL of filtered samples was injected into the LC/MS system, and a column BEH C18 (1.7 μm. 2.1 mm × 100 mm, Waters) was used for chromatographic separation, kept at a constant temperature of 45 °C. The mobile phases were prepared with 5 mM ammonium formate and 0.1% formic acid in water (eluent A) and in methanol (eluent B). The gradient elution was as follows: 95% eluent A for 0.25 min, followed by a shift to 5% eluent A until 9 min. The gradient was then returned to the initial condition of 95% eluent A at 9.51 min and maintained until 12 min. Pesticide residues were detected with an electrospray ionization source (ESI) set up in positive ionization mode using multiple reaction monitoring (MRM). The mass spectrometer operational mode was as follows: Capillary Voltage, 2 kV. Source Offset, at 30 V. Source Temperature, 150 °C. Desolvation Temperature, 600 °C. Cone Gas Flow, 199 L/h. Desolvation Gas Flow, 1000 L/h. Collision Gas Flow, 0.14 mL/min. Nebulizer Gas Flow, 7.00 Bar. LM 1 Resolution, 2.8. HM 1 Resolution, 15. Ion Energy, 1 0.5. LM 2 Resolution, 2.8. HM 2 Resolution, 15. Ion Energy, 2 1.5. Gain, 1.00. The pesticide mass transitions used for quantification and identification confirmation are reported in Appendix A. The compounds were quantified using external calibration within a concentration range of 5–200 ng/L.

### 2.3. Polyphenolic Profile

Analysis of polyphenolic compounds of OOMW was performed with a Liquid Chromatography system (Ultimate 3000) coupled with a quadrupole mass Spectrometer TSQ Quantiva triple-stage (Thermo Fisher Scientific, San José, CA, USA). Samples OOMW-A and OOMW-B were prepared via centrifugation of 5 mL at 5000 rpm for 25 min followed by filtration (0.45 μm, 13 mm, CLARIFY-PTFE filters, Washington, USA) and dilution with methanol (100-fold) Then, 5 μL were used for the injection.

The identification of polyphenols was partially based on the method described by Indelicato et al. [16]. Specifically, chromatographic separation was performed with a Hypersil GOLD C18 reversed-phase analytical column (2.1 mm × 50 mm, 1.9 μm particle size, Thermo Fisher Scientific) maintained at 30 °C,. Mobile phase A (water with 0.1% formic acid) and B (methanol) were used at a constant flow rate of 300 μL/min. The gradient program started at 0–2 min, 5% B; after 2–10 min, it linearly increased to 70% B; 10–12 min, linear increase to 100% B; 12–17 min, hold at 100% B; 17.0–17.1 min, linear decrease to 1% B; 17.1–19 min, hold at 1% B.

The mass spectrometer (QqQ, Thermo Scientific, Bremen, Germany) was equipped with a heated electrospray ionization (HESI) source set up in negative ion mode and tuned using standard solutions of each analyte (1 ppm in methanol). The optimized mass spectrometry parameters were the following: HESI (-), spray voltage: 2500 V; auxiliary gas pressure: 10 a.u.; sheath gas pressure: 50 psi; sweep gas: 1 a.u.; ion transfer tube temperature: 325 °C; vaporizer temperature: 350 °C; dwell time: 100 ms; Q1 resolution: 1 Da; Q3 resolution: 0.4 Da; collision-induced dissociation (CID) gas (argon) pressure: 2.0 mTorr; injection volume: 5 μL.

Selected reaction monitoring (SRM) was performed on the deprotonated molecules for each polyphenol ([M-H]-), with SRM transitions detailed in Appendix A. Quantification was based on the integration of peak areas for all monitored transitions. Pure standards were used for quantification, including Apigenin 7-Glucoside, apigenin, Quercetin, gallic acid, L-Mandelic Acid, Chlorogenic Acid, hydroxycinnamic acid, Kaempferol, caffeic acid, vanillic acid, Catechin, Rutin, Coumaric Acid, syringic acid, Gentisic Acid, ferulic acid, luteolin, hydroxytyrosol, oleaceinic acid, oleocanthal.

For phenolic compound quantification, an external calibration method was employed. A methanolic solution containing 5 ppm of each standard was prepared, from which five calibration solutions with concentrations ranging between 1 ppm, 500 ppb, 250 ppb, 100 ppb, 50 ppb, and 5 ppb. The linear correlation coefficient (R^2^) for all compounds was on average 0.99. Data were analyzed using a Quan/Qual Browser Trace Finder (Thermo Fisher Scientific, San José, CA, USA). Each calibration point represented the average of three independent injections.

Limits of detection (LODs) and quantification (LOQs) for all analytes were determined using a blank signal and regression curve (five blank injections between standards), collected within the same elution time window as the target compounds. The LOD was defined as the concentration yielding a signal equivalent to the blank signal plus three times its standard deviation, while the LOQ was defined as the concentration yielding a signal equivalent to the blank signal plus ten times its standard deviation.

### 2.4. Microbiological Analysis

Each OOMW (10 mL) was suspended in 90 mL of Ringer’s solution (Oxoid, Basingstoke, UK) through magnetic stirring in a 250 mL Pyrex Erlenmeyer flask. The resulting cell suspensions were then serially diluted (1:10) in Ringer’s solution and plated on selective agar media as reported by Messina et al. [17]. Briefly, the following microbial populations were evaluated: total mesophilic microorganisms (TMMs); members of the Enterobacteriaceae family; coagulase-positive staphylococci (CPS); *Escherichia coli*; *Listeria monocytogenes*; *Salmonella* spp.; enterococci; pseudomonads; yeasts; molds; aerobic spore-forming bacteria; mesophilic and thermophilic rods and coccus LAB. Microbiological analyses were carried out in triplicate using media and supplements purchased from Oxoid.

### 2.5. Isolation, Typing, and Identification of LAB

Presumptive LAB (Gram-positive and catalase-negative) were isolated, purified, and microscopically investigated.

LAB strain typing was performed through the random amplification of polymorphic DNA (RAPD)-PCR analysis following the approach described by Stefańska et al. [18].

All different strains were identified at the species level through 16S rRNA gene sequencing as described by Weisburg et al. [19]. The amplicons were purified and processed at BMR Genomics s.r.l. (Padova, Italy). The sequences obtained were compared with those available in the GenBank/EMBL/DDBJ https://www.ncbi.nlm.nih.gov (accessed on 28 September 2024) and EzTaxon-e databases https://eztaxon-e.ezbiocloud.net (accessed on 28 September 2024).

### 2.6. Technological Characterization of LAB

The LAB isolates were evaluated for their acidification capacity and antimicrobial activity. Acidification kinetics were measured following the method described by Alfonzo et al. [20], using sterile semolina extract (SSE) as the growth medium.

The antimicrobial activity of LAB was initially assessed using the agar-spot deferred method (ASDM) described by Schillinger and Lücke [21] and modified by Corsetti et al. [22]. *Escherichia coli* ATCC25922, *L. monocytogenes* ATCC19114, *Salmonella* Enteritidis ATCC13076, and *Staphylococcus aureus* ATCC33862 were used as indicator strains for the active substances. Strains that tested positive in the ASDM were further evaluated using the well diffusion assay (WDA) according to Schillinger and Lücke [21]. The protein nature of the inhibitory compounds was confirmed by treating the supernatant with proteinase K, protease B, and trypsin [23] and incubating for 18 h at 37 °C. A clear zone around the wells indicated a positive reaction to the enzyme treatment. All tests (acidification and inhibitory) were performed in duplicate.

### 2.7. Statistical Analysis

Phenolic profile, plate counts, acidification kinetic and antimicrobial activity data were subjected to ANOVA analysis using the software XLStat version 7.5.2 for Excel (Addinsoft, New York, NY, USA). Pairwise comparisons were conducted using Tukey’s test (*p* < 0.05).

## 3. Results and Discussion

### 3.1. Determination of Micro-Contaminants

The assessment of the potential use of OOMW was performed by conducting a multiresidual analysis of pesticide residues in samples collected in this study, targeting 166 active compounds using UPLC-MS/MS. The results are presented in Table 1. Our analysis reveals significant differences between the two samples in terms of both pesticide concentration and variety. In OOMW-A, four active compounds were detected: the systemic fungicide azoxystrobin, the neonicotinoid insecticide imidacloprid, and two herbicides, bromacil and simazine. Although none of these individual substances exceeded the regulatory limit (0.1 μg/L for single pesticides) [24], the total pesticide concentration (5.7 μg/L) was greater than the permissible threshold for the sum of all pesticides, which was set at 0.5 μg/L [24]. In OOMW-B, we found a much higher level of contamination, with 16 active compounds detected and a total pesticide concentration of 65.8 μg/L, exceeding the EQS-MA threshold by more than two orders of magnitude. Several compounds were found at concerning levels, particularly fungicides such as Mandipropamid (10.6 μg/L) and Penthiopyrad (29.5 μg/L). Additionally, the presence of Fenpyrazamide (13.3 μg/L), a fungicide commonly used in viticulture to control *Botrytis*, is especially noteworthy. The detection of substances not specific to olive cultivation raises concerns about poor agricultural practices, suggesting indiscriminate and improper pesticide use. In addition, the stark difference in pesticide residues between the two samples underscores the heterogeneity of this matrix, which is heavily influenced by local agricultural practices. This variability highlights the importance of conducting comprehensive, site-specific evaluations of pesticide residues when assessing the safety of OOMW. Another concern relates to the detection of six substances banned in Europe according to EC Regulation 1107/2009 [25], with three found in OOMW-A and four in OOMW-B. For example, chlorpyrifos and imidacloprid were detected in OOMW-B at concentrations of 0.14 μg/L for chlorpyrifos-ethyl and 1.25 μg/L for imidacloprid. The latter was also found in OOMW-A at a concentration of 0.17 μg/L. Although chlorpyrifos has been prohibited due to its impact on human health [26,27,28,29] and imidacloprid because of its effects on non-target organisms, such as pollinators [30,31], they were the ten most frequently used active substances, contributing significantly to the annual pesticide load and application rates per hectare [32]. This may be attributed both to the environmental persistence of these compounds [32,33,34] and to European regulatory provisions that allow EU member states to issue emergency authorizations for specific active substances [35]. This underscores the urgent need for a more stringent regulatory framework and enhanced pesticide management practices. Although these findings raise significant environmental concerns, it is essential to clarify that the residues found in OOMW do not directly indicate potential contamination of olive products, as the accumulation of residues depends on the hydrophilic and hydrophobic properties of the detected compounds (Table 1).

However, the determination of pesticide in the OOMW samples reveals high levels of active compounds, rendering this matrix unsuitable for safe application in food product functionalization. This calls for a reassessment of the agricultural inputs used during cultivation and emphasizes the importance of considering the potential use of by-products from organic farming.

### 3.2. Phenolic Profile of OOMW

The evaluation of polyphenols in OOMW samples was performed using LC-MS/MS experiments targeting 23 compounds. Although OOMW is considered a pollutant by-product generated during the production of oil making, it is characterized by medium–high content of phenolic compounds, making it a promising candidate for potential applications in food products due to their significant antioxidant properties.

Phenolic extracts from OOMW have also been tested in food emulsions, dairy products, and other model systems, demonstrating promising results with minimal impact on sensory characteristics or other properties. Our goal is to optimize their application and improve the nutritional and technological quality of OOMW.

A targeted LC-MS/MS method was developed to analyze its bio-phenolic composition by focusing on the characteristic polyphenols of olive oil reported in the literature [36,37].

Olive oils contain various classes of bioactive compounds, including alcohols, flavonoids, phenolic acids, secoiridoid derivatives, lignans, and phenylpropanoids [38,39,40]. Hydroxytyrosol and tyrosol are the most abundant phenolic alcohols whose concentrations increase during olive oil storage as a result of secoiridoid hydrolysis. The primary secoiridoids found in olive fruits are oleuropein and ligstroside [41]. The dialdehydic forms derived from these secoiridoids are known as oleacein and oleocanthal [37,39]. Several factors influence the composition of phenolic alcohols and secoiridoids, such as the olive cultivar, climate, soil, irrigation, ripeness level, and the technical process used for oil extraction [39]. Phenolic acids include hydroxycinnamic acid and its derivatives (e.g., p-coumaric, caffeic, and ferulic acids), as well as hydroxybenzoic acid and its derivatives (e.g., gallic, syringic, vanillic, and protocatechuic acids). Minor constituents identified include flavones such as luteolin, diosmetin, and apigenin [40].

LC-MS/MS analyses revealed a high concentration of hydroxytyrosol, secoiridoid derivatives, phenolic acids, and flavones (Table 2).

As previously mentioned, the concentration and composition of OOMW differ across regions due to various factors, including olive variety, ripeness, and the technical process used for oil production. Consequently, the profile and concentration of phenolic compounds in OOMW vary significantly between studies. The absence of oleuropein in our samples, which is reported as the major phenolic compound by Schmidt et al. [41], could be attributed to the degree of ripeness (harvest period). At this stage, oleuropein and ligstroside would have already degraded into hydroxytyrosol and other derivative species, which explains their high levels quantified in OOMW extracts [42,43]. In other studies, vanillin, ferulic acid, and vanillic acid were reported as the major polyphenols in French and Tunisian OOMW extracts [44,45]. Dermeche et al. [46] identified additional polyphenolic compounds in Algerian OOMW, including phenolic acids (such as gallic acid, cinnamic acid, and vanillic acid), hydroxybenzoic acid derivatives (e.g., 4-hydroxybenzoic acid and protocatechuic acid), and other phenolic compounds like 4-methylcatechol and 3,4-dihydroxyphenylacetic acid.

However, it is important to note that only a small percentage (about 2%) of the total phenolic content of the olive fruit is transferred into the oil during the extraction and production process, while the majority is unfortunately lost in waste products (OOMW) [47]. Therefore, the recovery and quantification of phenolic content in these by-products is crucial for the functionalization of food products.

### 3.3. Microbiological Analysis

The microbiological analysis of OOMW samples, carried out using a culture-dependent approach, included the search for pro-technological, spoilage, and pathogenic microorganisms commonly found in fruit and vegetables [48,49]. This evaluation is essential before using plant-derived products in food applications, as their origin often exposes them to microbial contamination from soil, water, and air, making them potential vehicles for undesired microorganisms [50]. The results of the viable counts of the OOMW samples are reported in Table 3.

The levels of TMM, mesophilic coccus and rod, and thermophilic coccus and rod LAB were superimposable among samples analyzed (*p* > 0.05). Both OOMWs hosted levels of TMMs of about 10 CFU/mL and LAB at 10^2^ CFU/mL. Slightly higher levels were reported by El Yamani et al. [51] for olive mill wastewater from different regions of northern Morocco. Despite being present in low levels, the presence of LAB demonstrated their ability to withstand the polyphenolic compounds of the OOMW [52]. The specific search for spoilage (*Pseudomonas* spp., unicellular and filamentous fungi) and pathogenic (aerobic spore-forming bacteria, CPS, *L. monocytogenes*, members of the Enterobacteriaceae family, *E. coli*, and *Salmonella* spp.) microorganisms, commonly associated with poor hygiene of food productions [53], did not reveal their presence at detectable levels (<1 log CFU/mL) in either sample analyzed. These samples also showed undetectable levels of enterococci. These bacteria are part of the LAB group and contribute positively to the fermentation of animal-based products [54]. However, their presence in raw materials and processed foods poses a risk to consumers, as they can acquire and transfer genes responsible for antimicrobial resistance and virulence traits [55]. These results highlight the microbiological suitability of OOMW for food production.

### 3.4. Phenotypic and Genotypic Characterization of LAB

In total, 12 colonies of presumptive LAB (Gram-positive and catalase-negative) distinguished into 8 cocci and 4 rods were isolated from both OOMW samples. These bacteria were purified and subjected to RAPD-PCR analysis, a technique widely used for the strain typing of LAB strains of food origin [56]. The agarose gel in Figure 1 shows the presence of only two distinct strains.

The sequencing of the 16S rRNA gene indicated that the LAB community resistant to polyphenols present in OOMW was represented by the species *Lactiplantibacillus plantarum* OMW 1 (Ac. No. PQ394646) and *Leuconostoc mesenteroides* OMW 23 (Ac. No. PQ394647). These LAB species are typical of olive products microbiota [57] and represent the most common species used as starter cultures in the production of fermented table olives [58].

### 3.5. Technological Traits of LAB

The acidification kinetics of the LAB resistant to polyphenols present in OOMW are reported Figure 2.

The two strains (*Lpb*. *plantarum* OMW1 and *Ln. mesenteroides* OMW23) showed high acidification capabilities, reaching pH values below 5.0 after 8 h of fermentation. At 24 h, both strains acidified the medium to below pH 4.0. The rapid acidification of LAB is an important criterion of selection for future applications in fermented food products [59]. Furthermore, LAB contribute to the prolonged shelf life of this product through many activities [60]. Among them, the production of active substances is related to secondary metabolism [61]. Bacteriocins are low-molecular-mass peptides or proteins (usually 30–60 amino acids) exerting a bactericidal or bacteriostatic effect that are synthesized at a ribosomal level and extracellularly released [62]. Thus, their basic characteristic is the protein structure [63]. The two LAB strains identified produced antibacterial compounds (Figure 3).

*Lpb. plantarum* OMW1 inhibited all four indicator strains, while *Ln. mesenteroides* OMW23 inhibited only *E. coli* and *S. enteritidis*. All active substances had a protein structure because the inhibition disappeared after proteolytic enzyme treatment. Although the protein nature is a characteristic of the inhibitory compounds ascribable to the bacteriocins, the activities found in the supernatants from the LAB characterized were referred to as bacteriocin-like inhibitory substances (BLIS) because the amino acid sequencing necessary to undoubtedly recognize them as bacteriocins [64] was not performed. These findings suggest that the strains isolated from OOMW are promising candidates for developing starter cultures to be used for food fermentation.

## 4. Conclusions

Agricultural and food industry W&BP are rich sources of bioactive compounds. However, extracting and purifying these substances can be costly and generate additional waste. Therefore, innovative recycling methods are needed to achieve two goals: prevent extra waste and add value to existing waste. One of the latest advancements in recycling W&BP, which is high in biologically active substances, is the production of functional foods. In the case of OOMW, there are no direct examples of its use as a food ingredient. This study found that OOMW samples contain significant levels of polyphenols, particularly hydroxytyrosol, secoiridoid derivatives, phenolic acids, and flavones. Additionally, microbiological analysis of the samples revealed low levels of viable bacteria, with food-grade LAB such as *Lpb. plantarum* and *Ln. mesenteroides* being identified. These LAB strains were also capable of inhibiting some foodborne pathogens, suggesting potential use in food biopreservation. However, the presence of residues from insecticides, herbicides, and fungicides makes OOMW unsuitable for food production. Given that pesticide residues are clearly a result of OOMW originating from conventionally farmed olives, future research will concentrate on characterizing and utilizing OOMW from organically farmed olives for the production of health-oriented sourdough bakery products.

## Figures and Tables

**Figure 1 foods-14-00449-f001:**
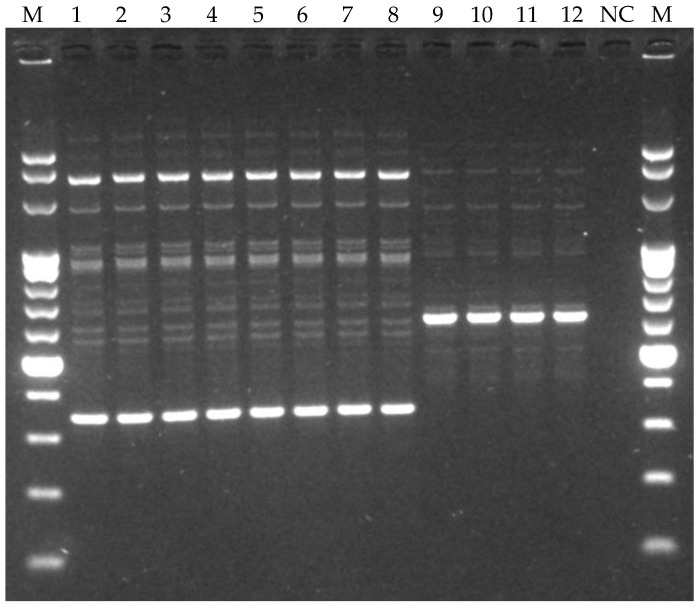
Image of agarose gel showing RAPD profiles of LAB resistant to olive oil mill wastewater. Lanes M, GeneRuler 100 bp DNA ladder (EURx-Molecular Biology Products, Gdansk, Poland); lanes 1–8, coccus LAB strains; lanes 9–12, rod LAB strains; lane NC, negative control.

**Figure 2 foods-14-00449-f002:**
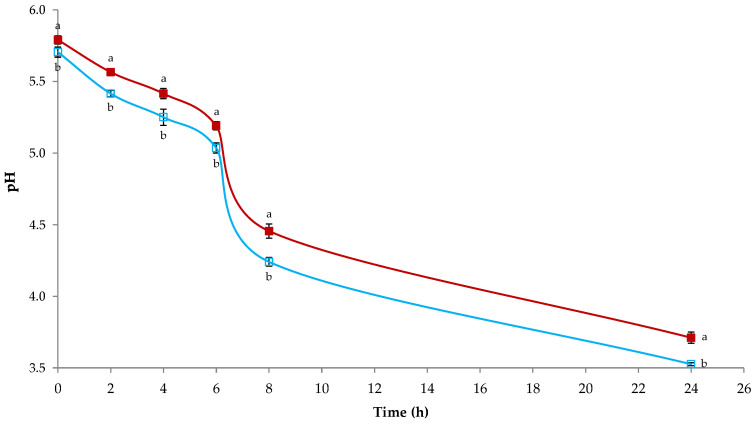
Kinetics of acidification of lactic acid bacteria resistant to olive oil mill wastewater. Blue line, *Lactiplantibacillus plantarum* OMW1; red line, *Leuconostoc mesenteroides* OMW23. Results indicate mean value ± S.D. (standard deviation) of four determinations. a, b = *p* < 0.05.

**Figure 3 foods-14-00449-f003:**
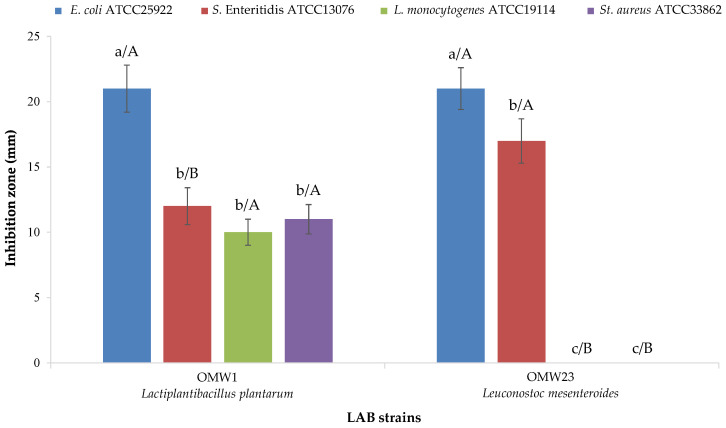
Bacteriocin-like inhibitory activity of lactic acid bacteria resistant to olive oil mill wastewater against the main foodborne pathogenic bacteria. Abbreviations: *E*., *Escherichia*; *S*., *Salmonella*; *L*., *Listeria*; *St*., *Staphylococcus*, LAB, lactic acid bacteria. a, b, c = *p* < 0.05; A, B = *p* < 0.05.

**Table 1 foods-14-00449-t001:** Active substances measured in olive oil mill wastewater samples and their physico-chemical properties according to the Pesticide Properties DataBase, University of Hertfordshire. (Available online: https://sitem.herts.ac.uk/aeru/ppdb/ (accessed on 11 November 2024).

Active Substances	OOMW-A (μg/L)	OOMW-B (μg/L)	logP (Kow)	Water Solubility	Fat Solubility	EC Regulation 1107/2009 Status	Mode of Action
Acetamiprid	n.d	0.11	0.8	High	Insoluble	Approved	Neonicotinoid insecticide
Azoxystrobin	0.02	0.38	2.5	Low	Insoluble	Approved	Fungicide
Bromacil	5.29	n.d	1.88	High	Not Reported	Not Approved	Herbicide
Chlorantraniliprole	n.d	1.19	2.86	Low	Soluble	Approved	insecticide
Chlorpyrifos-Ethyl	n.d	0.14	4.7	Low	Soluble	Not Approved	insecticide
Cyproconazol	n.d	0.71	3.09	Moderate	Soluble	Not Approved	Fungicide
Fenpyrazamine	n.d	13.32	3.52	Low	Soluble	Approved	Fungicide
Imidacloprid	0.17	1.25	0.57	High	Insoluble	Not Approved	Neonicotinoid insecticide
Mandipropamid	n.d	10.60	3.2	Low	Insoluble	Approved	Fungicide
Metalaxyl	n.d	4.75	1.75	High	Not Reported	Approved	Fungicide
Penthiopyrad	n.d	29.54	4.62	Low	Insoluble	Approved	Fungicide
Prometryn	n.d	0.12	3.34	Low	Not Reported	Not Approved	Herbicide
Pyraclostrobin	n.d	0.34	3.99	Low	Soluble	Approved	Fungicide
Simazine	0.21	n.d	2.3	Low	Not Reported	Not Approved	Herbicide
Tebuconazole	n.d	2.1	3.7	Low	Likely to be Soluble	Approved	Fungicide
Terbuthylazine	n.d	0.17	3.4	Low	Soluble	Approved	Herbicide
Terbuthylazin-Desethyl	n.d	0.34	2.3	Moderate	Not Reported	Not Reported	metabolite
Tetraconazole	n.d	0.68	3.5	Moderate	Soluble	Approved	Fungicide
∑pesticide	5.7	65.8					

Abbreviation: OOMW, olive oil mill wastewater; Kow, octanol–water partition coefficient at pH 7, 20 °C; n.d, not detectd.

**Table 2 foods-14-00449-t002:** Polyphenolic compounds detected in olive oil mill wastewater samples.

Active Substances	Samples		*p* Value
	OOMW-A (μg/mL)	OOMW-B (μg/mL)	
Caffeic acid	40.1 ± 2.4 a	25.3 ± 1.4 b	0.001
Hydroxycinnamic acid	92.2 ± 3.7	88.5 ± 3.9	0.298
Hydroxytyrosol	143.0 ± 5.1	148.0 ± 4.3	0.266
Cumaric acid	71.6 ± 3.2	77.9 ± 4.0	0.100
Ferulic acid	20.9 ± 1.2 a	12.7 ± 0.6 b	0.001
Oleaceinic acid	8.1 ± 0.5	8.2 ± 0.5	0.815
Luteonin	9.9 ± 0.4 b	10.8 ± 0.3 a	0.040
Apigenin	6.0 ± 0.3	6.1 ± 0.3	0.707
Σpolyphenols	391.8	377.5	

Abbreviations: OOMW, olive oil mill wastewater. On the row: a, b = *p* < 0.05.

**Table 3 foods-14-00449-t003:** Microbial loads (CFU/mL) of olive oil mill wastewater samples.

Microorganisms	Samples		*p* Value
	OOMW-A	OOMW-B	
TMM	1.85 ± 0.34	1.7 ± 0.29	0.683
Mesophilic coccus LAB	2.40 ± 0.23	2.33 ± 0.19	0.491
Thermophilic coccus LAB	2.35 ± 0.43	2.23 ± 0.39	0.799
Mesophilic rod LAB	2.32 ± 0.23	2.43 ± 0.32	0.654
Thermophilic rod LAB	1.70 ± 0.54	2.43 ± 0.49	0.271
Enterobacteriaceae	<1	<1	n.e.
Enterococci	<1	<1	n.e.
CPS	<1	<1	n.e.
*L. monocytogenes*	<1	<1	n.e.
*E. coli*	<1	<1	n.e.
*Salmonella* spp.	<1	<1	n.e.
Pseudomonads	<1	<1	n.e.
Aerobic spore-forming bacteria	<1	<1	n.e.
Yeasts	<1	<1	n.e.
Molds	<1	<1	n.e.

Abbreviations: OOMW, olive oil mill wastewater; LAB, lactica acid bacteria; TMM, total mesophilic microorganisms; CPS, coagulase-positive staphylococci; *L.*, *Listeria*; *E.*, *Escherichia*; n.e., not evaluated.

## Data Availability

The original contributions presented in this study are included in the article and Appendix A. Further inquiries can be directed to the corresponding author.

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
