# Peer review of "Analysis of Olive Oil Mill Wastewater from Conventionally Farmed Olives: Chemical and Microbiological Safety and Polyphenolic Profile for Possible Use in Food Product Functionalization"

_foods, 2025, doi:10.3390/foods14030449_

Round 1
Reviewer 1 Report
Comments and Suggestions for Authors
The comments are in the attached file.

Author Response
AU. Thank you for your feedback and for offering valuable comments to enhance the manuscript's quality. We have taken all your suggestions into account, and the changes have been highlighted in yellow in the text.
1_In Abstract section: It is not clear what is the difference between two samples – OOMW-A and OOMW-B.
AU. We agree with your comment. This missing information has been added to the text (L17-18). Please note that the maximum word count allowed for abstracts by Foods is 200 words and for this reason more details is given in M&M section at the paragraph 2.1.
2_In Introduction section: I recommend to deepen the information about the production of functional food products enriched with olive oil wastes in order to respond to the title of the article.
AU. We appreciate the reviewer constructive comment. The requested information have been added in the text (L67-72) to encounter your request.
3_In Results and discussion section (page 7, Table 2): The standard deviations of polyphenolic compounds are missing.
AU. We completely agree with your suggestion. Standard deviations have been added in Table 2 (L349).
4_In Results and discussion section (page 7, Table 2): I recommend to add a test of significant difference between two samples (OOMW-A and OOMW-B) in this Table.
AU. We completely agree with your suggestion. Changes have been made in the Table 2 (L349).
5_In Results and discussion section (Figures 2 and 3): I recommend to add a test of significant difference between two samples presented in Figures 2 and 3.
AU. We completely agree with your suggestion. Changes have been made in the Figure 2 and 3 (L427; 443).
6_In Conclusions section: It is good to explain what future food processing applications are envisaged
AU. We completely agree with this comment. Additional details on future research have been reported in the conclusions section (L466-469).
Reviewer 2 Report
Comments and Suggestions for Authors
Comments and Suggestions for Author
Manuscript Number:foods-3396648
The manuscript entitled “Analysis of olive oil mill wastewater from conventionally farmed olives: chemical and microbiological safety and polyphenolic profile for possible use in food product functionalization” studied aimed to perform an in-depth investigation of olive oil mill wastewater (OOMW) and the primary objective of this study was to analyze OOMW for its potential application in food production.However, the study does not provide clear guidance on how to efficiently separate and utilize the beneficial substances in OOMW, making it less practical.
Generally, this work is recommended for publication in foods after a major revision.
Some of the major points to be answered are listed below.
1. At the end of the abstract, you can briefly mention the direction of future research or the significance of this research, such as how to solve the problem of wastewater pollutants, how to further validate the function of lactic acid bacteria, etc.
2. In the introductory section, the environmental challenges of OOMW in the olive oil industry could be further clarified and refined by specifying the common methods currently used to deal with OOMW and their limitations. This would better draw out the innovation and necessity of your research.
3. This article focuses on analyzing the beneficial components in OOMW but does not address their isolation and utilization. I suggest the authors include experimental or more effective evidence on how to isolate and utilize these components to enhance the practical value of the study.
4. Many of the references in the article are outdated. It is recommended that references from the last five years be cited and referenced to make the article more convincing.
Author Response
AU. Thank you for your feedback and for offering valuable comments to enhance the manuscript's quality. We have taken all your suggestions into account, and the changes have been highlighted in green in the text.
1_At the end of the abstract, you can briefly mention the direction of future research or the significance of this research, such as how to solve the problem of wastewater pollutants, how to further validate the function of lactic acid bacteria, etc.
AU. We completely agree with this comment. Additional details on future research have been reported in the abstract (L33-35).
2_In the introductory section, the environmental challenges of OOMW in the olive oil industry could be further clarified and refined by specifying the common methods currently used to deal with OOMW and their limitations. This would better draw out the innovation and necessity of your research.
AU. We appreciate the reviewer constructive comment. The requested information have been added in the text (L51-59) to encounter your request.
3_This article focuses on analyzing the beneficial components in OOMW but does not address their isolation and utilization. I suggest the authors include experimental or more effective evidence on how to isolate and utilize these components to enhance the practical value of the study.
AU. Thank you for this comment, which gives us the opportunity to clarify the aim of this study. The aim was not to isolate and utilise the bioactive compounds present in OOMW, but to analyse this waste for chemical and microbiological safety and to evaluate the polyphenol profile for use as is in fermented functional foods. In fact, Line 70-71 specifies that information on the direct use of OOMW is still limited. This aspect has been highlighted in the introduction for greater clarity (L75-83).
4_Many of the references in the article are outdated. It is recommended that references from the last five years be cited and referenced to make the article more convincing.
AU. We perfectly agree with you. Sixteen autdated references have been changed in the text and in the references list.
Round 2
Reviewer 2 Report
Comments and Suggestions for Authors
Manuscript Number:foods-3396648
The manuscript entitled “Analysis of olive oil mill wastewater from conventionally farmed olives: chemical and microbiological safety and polyphenolic profile for possible use in food product functionalization” studied aimed to perform an in-depth investigation of olive oil mill wastewater (OOMW) and the primary objective of this study was to analyze OOMW for its potential application in food production. Firstly, I would like to express my appreciation to the authors for their careful consideration of my previous comments and for the revisions they have made. Then, I had previously focused on the practicality of the later and ignored the author's real purpose was not to isolate and utilise the bioactive compounds present in OOMW, but to analyse this waste for chemical and microbiological safety and to evaluate the polyphenol profile for use as is in fermented functional foods.After reviewing the revised manuscript, I believe the authors have addressed my concerns effectively, which have significantly improved the clarity and quality of the paper. I have no further comments or concerns, so I have no further suggestions to make.